# A Light-Weight Network for Small Insulator and Defect Detection Using UAV Imaging Based on Improved YOLOv5

**DOI:** 10.3390/s23115249

**Published:** 2023-06-01

**Authors:** Tong Zhang, Yinan Zhang, Min Xin, Jiashe Liao, Qingfeng Xie

**Affiliations:** College of Mechanical and Electrical Engineering, Guilin University of Electronic Technology, Guilin 541004, China; mala@guet.edu.cn (T.Z.); xinmin1203@163.com (M.X.); 20012201020@mails.guet.edu.cn (J.L.); 18966409631@163.com (Q.X.)

**Keywords:** keyword light-weight, insulator and defect detection, YOLOv5, ghost module, convolutional block attention module, unmanned aerial vehicles

## Abstract

Insulator defect detection is of great significance to compromise the stability of the power transmission line. The state-of-the-art object detection network, YOLOv5, has been widely used in insulator and defect detection. However, the YOLOv5 network has limitations such as poor detection rate and high computational loads in detecting small insulator defects. To solve these problems, we proposed a light-weight network for insulator and defect detection. In this network, we introduced the Ghost module into the YOLOv5 backbone and neck to reduce the parameters and model size to enhance the performance of unmanned aerial vehicles (UAVs). Besides, we added small object detection anchors and layers for small defect detection. In addition, we optimized the backbone of YOLOv5 by applying convolutional block attention modules (CBAM) to focus on critical information for insulator and defect detection and suppress uncritical information. The experiment result shows the mean average precision (mAP) is set to 0.5, and the mAP is set from 0.5 to 0.95 of our model and can reach 99.4% and 91.7%; the parameters and model size were reduced to 3,807,372 and 8.79 M, which can be easily deployed to embedded devices such as UAVs. Moreover, the speed of detection can reach 10.9 ms/image, which can meet the real-time detection requirement.

## 1. Introduction

Insulators in power systems play an irreplaceable role in the mechanical support and electrical insulation of transmission lines. Most insulators in transmission lines operate outdoors and are subject to various environmental interferences [1]. In the event of insulator failure or explosion, it can affect the power supply and cause significant damage to the power transmission system [2].

In recent years, the self-explosion and branch-drop of insulators have become a common problem in high-voltage transmission lines. It takes much work to check for insulator defects manually. Therefore, UAVs have become increasingly common to capture insulator images and identify defective insulators by deploying the object detection algorithm on embedded devices [3,4].

We can divide object detection methods into two types [5]. One type is the traditional object detection method, such as the Support Vector Machine (SVM) [6], Viola Jones (VJ) detector [7], Deformable Parts Mode (DPM) detector [8], and Histogram of Oriented Gradients (HOG) detector [9]. However, traditional object detection algorithms that select features largely depend on manual work, and in dealing with challenges such as a complex background and significant camera dynamics, these methods have poor robustness and low accuracy [10].

Another type of object detection method is based on deep learning, including one-stage and two-stage algorithms. One-stage algorithm extracts feature maps through a convolution neural network to infer the classification and bounding boxes of the target. Two-stage algorithm first extracts region proposals and infers the classification and location of the target through the convolution network.

Girshick et al. first proposed R-CNN [11] in 2014, which is a two-stage algorithm, it applied a convolution neural network to extract features, and with the excellent performance of CNN, the accuracy from the PASCAL VOC dataset is raised from 35.1% to 53.7%. Then they proposed Fast R-CNN [12] and Faster R-CNN [13] to improve detection performance. Wang et al. [14] proposed a method with improved Faster R-CNN combined with the backbone of Resnet 50 for insulator detection. However, its low detection speed and large number of parameters cannot meet the real-time detection requirement.

In contrast, Redmon et al. proposed the one-stage algorithm You Only Look Once (YOLO) [15] series, which are mainstream object detection algorithms. Compared with the R-CNN series, the YOLO series are more suitable for applications of real-time detection. Yang et al. [16] proposed an improved YOLOv3 insulator detection method. This method added additional max pooling layers and Smooth-EIoU Loss, and the mAP reached 91.5%. Zhang et al. [17] proposed an insulator and defect dataset with synthetic Fog and an algorithm based on YOLOv5 with channel attention mechanisms and SE-CSP layer to effectively distinguish the features. The mAP0.5 and mAP0.5:0.95 reached 99.5% and 88.3%. These methods have great significance to insulators and defect detection, but there remain some problems. (1) These methods have high detection accuracy on large objects such as insulators but low detection rates on small defects. (2) Most of the methods have a large number of parameters and floating-point operations (FLOPs), large model sizes, and low detection speed. (3) Some light-weight models can efficiently reduce the size and parameters of the model, but they cannot achieve a balance between model size and accuracy.

Therefore, this paper proposed a light-weight insulator and defect detection network based on YOLOv5s, which can maintain high detection speed and accuracy. Our contributions are as follows:A ghost module [18] is introduced into the YOLOv5 backbone and neck to reduce the parameters and model size.A small object detection network is proposed to increase the detection rate of small insulator defects.CBAM [19] is applied to the backbone of the network to select critical features of insulators and defects and suppress the uncritical features to improve the accuracy of the network.Compared with the methods in the literature, we reduced the network parameters and model size, which can make our method deploy into embedded devices such as UAVs and reduce costs. Our small object detection network makes it easier to detect small defects which the literature methods cannot detect. Our method has a smaller model size and higher accuracy than other methods mentioned in the literature.

## 2. The Architecture of Original Network

YOLOv5, proposed by Glenn Jocher, was a state-of-the-art object detection algorithm. This one-stage algorithm is more suitable for real-time detection in UAVs than a two-stage algorithm. The YOLOv5 network structure includes a backbone, neck, and head, as shown in Figure 1.

### 2.1. Backbone

The improved CSP-Darknet53 is used in YOLOv5 as the backbone. Darknet53 refers to the structure of ResNet and adds residual blocks to the network architecture. It increases the network depth while controlling the gradient propagation and avoiding gradient exploding or vanishing. The C3 block was proposed to replace the original CSP block. It removed the Conv block after the residual output and changed LeakyRelu to SiLU [20] to reduce the FLOPs.

### 2.2. Neck

The network structure of Neck follows the structure of FPN (Feature Pyramid Network) [21] with PAN (Path Aggregation Network) [22] to combine and transfer features in local (low-level) and global (high-level). The feature map carries richer semantic features and weaker localization features on a higher level, and it takes richer localization features and weaker semantic features on a lower level. The FPN helps combine the rich semantic features from higher to lower levels. In contrast, PAN transmits rich localization features from lower to higher levels. With the combination of PAN and FPN, the parameters are aggregated from the backbone layer and improve the feature extraction capability.

### 2.3. Head

The head has three detection layers which make predictions based on the feature maps generated by the neck. It has three detection layers consisting of three sizes of feature maps for different sizes of the object. It outputs the bounding boxes, the classification of the object, and the confidence score.

Figure 1 shows the original network of YOLOv5, the network structure of the backbone, which in the blue box includes the Conv layer, C3 layer, and SPPF layer. The Conv layer extracts feature maps through convolution operation, normalizes the distribution of eigenvalues, and introduces nonlinear transformation capability to realize the transformation and extraction of input features. The C3 layer improves the ability of feature extraction by increasing the depth of the network and the receptive field. The SPPF layer can convert feature maps of any size into fixed-size feature vectors. The neck network structure in the orange box includes the Conv layer, C3 layer, Concat layer, and Upsample layer. The Concat layer concatenates two or more feature maps in the channel dimension. The Upsample layer is to upsample the input feature map and expand the scale of the feature map. The head network structure, which is in the purple box, includes three Conv2d layers. The Conv2d layer only extracts features through convolution operation.

## 3. Method

### 3.1. Improved YOLOv5 Method

The pretrained weights include YOLOv5n, YOLOv5s, YOLOv5m, YOlOv5l and YOLOv5x. In comparison, the mAP of YOlOv5s is lower than YOLOv5x, but the parameters and FLOPs are much fewer than YOLOv5x, and it is more adaptive for deployment to embedded devices such as UAVs. Therefore, we chose YOLOv5s as the pretrained weights for our model. Besides, we selected the latest version, YOLOv5v6.2, as the benchmark. The Focus Layer was replaced with an equivalent 6 × 6 convolutional layer, and the latter has better exportability. Besides, Spatial Pyramid Pooling-Fast (SPPF) layer was proposed to replace the Spatial Pyramid Pooling (SPP) [23] layer. The SPP layer contains three different max-pooling layers in parallel connection, as shown in Figure 2, to avoid shape and incomplete cropping distortion of images caused by the R-CNN algorithm. Moreover, it solves the problem of multiple feature extraction to improve the speed of generating bounding boxes. The MaxPool2d layer extracts the maximum feature map of the specified kernel size and significantly reduces the feature map size.

The architecture of the SPPF layer is connected by three max-pooling layers in sequence, which can reduce the computing cost, as shown in Figure 3. Moreover, SPPF can perform the same functions as SPP in half the time. The ConvBNSiLU layer plays the same role as the Conv layer in the backbone. It extracts feature maps through convolution operation, normalizes the distribution of eigenvalues, and introduces nonlinear transformation capability to realize the transformation and extraction of input features.

### 3.2. Lightweight Network with Ghost Module

The Ghost module is a model compression method. It can extract more feature maps with a smaller number of parameters to raise the computing speed and reduce latency in UAVs. The working process of the Ghost module is shown in Figure 4. Figure 4a is the process of the original convolutional layer. Figure 4b is the process of the Ghost module. The Ghost module first generates intrinsic feature maps through original convolution; then generates ghost feature maps through some cheap transformation operations ($\Phi_{i}$). Finally, it concatenates the intrinsic feature maps and ghost feature maps for output.

From the original convolutional layer, the input feature map is h×w×c, and the input kernel size is k×k, where h is the input feature map height, w is the input feature map width, and c is the input channel numbers. It outputs the h’×w’×n feature map, where h’ is the output feature map height and w’ is the output feature map width, and n is the output channel numbers. During the original convolution process, FLOPs numbers are required to be computed as n·h’·w’·c·k·k, in which the number of filters n is often over $10^{5}$ and the number of channels c is 256 or 512. The Ghost module extracts ghost feature maps by applying some cheap transformation operations and generates some intrinsic feature maps by using fewer filters. In Ghost module computation, the FLOPs can be computed as $\left(s-1\right)\cdot n/s\cdot h’\cdot w’\cdot d\cdot d+n/s\cdot h’\cdot w’\cdot c\cdot k\cdot k$, the d×d is equal to each linear operation averaged kernel size and s is the cheap transformation operation numbers and s is much smaller than c.
(1)$$\begin{array}{c} r_{s}=\frac{h’\cdot w’\cdot c\cdot k\cdot k\cdot n}{(s-1)\cdot \frac{n}{s}\cdot h’\cdot w’\cdot d\cdot d+\frac{n}{s}\cdot h’\cdot w’\cdot c\cdot k\cdot k}\\ =\frac{c\cdot k\cdot k}{\frac{1}{s}\cdot c\cdot k\cdot k+\frac{s-1}{s}\cdot d\cdot d}\approx \frac{s\cdot c}{s+c-1}\approx s \end{array}$$

From Formula (1), the FLOPs required for the original convolutional layer are s times that of the Ghost module, and the parameters for computation also can be approximately equal to s. This shows the advantages of the Ghost module in a lightweight network. Therefore, GhostBottleneck and C3Ghost are proposed to further optimize network parameters [24]. Figure 5 shows the architecture of GhostBottleneck and C3Ghost. GhostBottleneck is designed in place of the bottleneck in the original convolution, and C3Ghost is designed to replace the C3 layer. We replaced all original convolutional layers with Ghost convolutional layers. We replaced all C3 layers with C3Ghost layers to reduce the FlOPs and computation parameters to better meet the real-time detection requirement and make it easier to deploy to UAVs.

### 3.3. CBAM Attention Mechanism

To make the algorithm focus on the most critical information through the inputs and decrease the attention to other useless information, we introduce CBAM to our network.

CBAM is a widely used and lightweight attention mechanism, which can be divided into CAM (channel attention module) and SAM (spatial attention module). When inputting the feature map $F\in R^{C\times H\times W}$, The CBAM infers one-dimensional channel attention feature map $M_{C}\in R^{C\times 1\times 1}$, and two-dimensional spatial attention feature map $M_{S}\in R^{1\times H\times W}$. The process of CBAM can be illustrated as:(2)$$F^{\prime }=F⊗M_{c}^{}(F)$$
(3)$$F^{\prime \prime }=F^{\prime }⊗M_{s}(F^{\prime })$$
where *C* is the feature map channel, *H* is the feature map height, *W* is the feature map width and ⊗ is element multiplication. $F^{\prime \prime }$ is the final output.

The channel attention focuses on key channels by multiplying weights from different channels. The channel attention map $M_{C}\in R^{C\times 1\times 1}$ is computed as:(4)$$M_{c}(F)=\sigma (W_{1}(W_{0}(F_{avg}^{c}))+W_{1}(W_{0}(F_{max}^{c})))$$
where *σ* is the sigmoid activation function, $W_{1}$ and $W_{0}$ are the weights that share their inputs, and $F_{avg}^{c}$ and $F_{max}^{c}$ are the features of the average-pool and the features of the max-pool.

Spatial attention learns spatial features by focusing on location information and assigning weights for spatial features. The spatial features can be calculated as:(5)$$M_{s}(F)=\sigma (f^{7\times 7}([F_{avg}^{s};F_{max}^{s}]))$$
where $f^{7\times 7}$ is the convolution with 7×7 filter size.

As shown in Figure 6, we introduced CBAM to the backbone of our improved network structure to focus on insulator and defect information, suppress other useless information and increase the detection rate and efficiency of insulator detection.

### 3.4. Small Object Detection Network

In the pictures taken by UAVs, the proportion of insulators and defects in the image is very small. We changed the backbone and neck structure to improve the small object detection accuracy. The head network is aimed to predict classes and bounding boxes from the feature maps extracted from the neck. It has little impact on small object detection. So, we only increased the number of detection heads.

In the backbone, we redirected the connection of bottom layers, which include feature maps in higher resolution, to send these feature maps directly into the neck. Moreover, we expanded the neck to adapt to extra feature maps. The structure of our network is shown in Figure 7. In addition, we changed the sizes and numbers of the anchors applied in the head and changed the parameters to be adjusted to this new structure.

In Figure 7, we replaced most of the Conv layers with the GhostConv layer and replaced the C3 layers with the C3Ghost layers to reduce parameters. Besides, we added CBAM layers on the bottom of the C3Ghost layers in the backbone improve the accuracy. Then, we applied our small object detection network which is in the green box to the bottom of the second C3Ghost layer in the neck and concat the first CBAM layer in the backbone. We applied another Conv2d layer in the head connected with the third C3Ghost layer to improve the capability of small object detection.

## 4. Experiments

### 4.1. Experiment Introduction

Our dataset used three public insulator datasets: the Chinese Power Line Insulator Dataset (CPLD), provided by the State Grid Corporation of China, the Synthetic foggy insulator dataset (SFID), made by Zheng-De Zhang and one insulator dataset from the Internet.

The CPLD dataset contains 600 insulator images and 248 insulator images with defects partly collected by UAVs and partially generated by the data augmentation method. Moreover, the SFID was based on CPLD and Unifying Public Datasets for Insulators (UPID). It constructed the dataset by applying a synthetic fogging algorithm containing 13,718 insulator and defect images. The dataset from the internet includes 678 insulator and defect images. We combined and filtered these three types of insulator datasets. Our dataset contains 7582 images, and we divided the dataset into 7:2:1; 5307 images for training, 1517 images for verification, and 758 images for testing. The pixels of images are set into 1152 × 864 and stored in JPG format, and the labeling files are saved in VOC format.

All experiments were conducted on a Linux system, the CPU is AMD Ryzen 5, and the GPU is NVIDIA RTX2080Ti with 11 G memory. The software environment contains CUDA 11.2, CUDNN 8.40, Python 3.10, and PyTorch framework.

To test the performance of our proposed small insulator and defect detection model, we chose precision, recall, mAP0.5, mAP0.5:0.95, FLOPS, parameters, and model size as evaluation indicators.

Precision is the proportion that the model predicted positive sample numbers correctly. It can be defined as:(6)$$Precision=\frac{T_{P}}{T_{P}+F_{P}}$$

The recall is the proportion that the model correctly predicted. It can be defined as:(7)$$Recall=\frac{T_{P}}{T_{P}+F_{N}}$$ where the $T_{p}$ is the positive sample number, the model predicted it as positive. $F_{P}$ is the positive sample number the model predicted it as positive. $F_{N}$ are the positive sample numbers the model ignored.

Intersection over Union (IoU) is the intersection and union ratio between bounding boxes and ground truth. It can be defined as:(8)$$IoU=\frac{T_{P}}{F_{P}+T_{P}+F_{N}}$$

When two bounding boxes overlap, the IoU can be set to 1. When two bounding boxes do not intersect, the IoU can be set to 0. Regardless of the size of prediction bounding boxes and the ground truth, the output IoU is always set between 0 and 1. Therefore, it can truly reflect the detection results.

Average Precision (*AP*) is equal to the area under the Precision-Recall curve, and it can be defined as:(9)$$AP=\int_{0}^{1}P(R)dR$$

Mean Average Precision (mAP) is the average precision of all classes, and it can measure the model’s performance across all classes. The formula of *mAP* can be defined as:(10)$$mAP=\frac{\sum_{i=1}^{N}AP_{1}}{N}$$
where mAP0.5 is the mean average precision when IoU is set to 0.5, and mAP0.5:0.95 is the mean average precision when IoU is set from 0.5 to 0.95 with 0.05 steps.

### 4.2. Experiment on Insulators and Defects

#### 4.2.1. Experiment Results

We chose several experiments to verify the model’s performance in different network structures. There are four experiments trained in this article, YOLOv5s, YOLOv5s + Ghost module, YOLOv5s + Ghost module + small object detection network, YOLOv5s + Ghost module + small object detection network + CBAM. Table 1 shows the training results in different network structures. Table 2 shows the deployment performance for UAVs. We trained 300 epochs, and Figure 8 shows the training cure of our final model in mAP0.5 and mAP0.5:0.95.

#### 4.2.2. Influence of Ghost Module

To reduce the model size and increase the performance of our model for UAVs, we introduced the Ghost module to the network structure. In Table 2, the parameters of the model with the Ghost module are reduced by 47.51%, and the FLOPs are reduced by 48.75% compared with YOLOv5s. The model size is reduced by 45.77%, and the GPU processing speed is slightly improved. However, the average accuracy decreases by 0.3% in the evaluation of mAP0.5 and 1.3% in the evaluation of mAP0.5:0.95 in Table 1. For insulators, the mAP0.5 reduces by 0.3%, and mAP0.5:0.95 decreases by 1.1%. For defect, the mAP0.5 decreases by 0.2%, and mAP0.5:0.95 decreases by 1.5%. Ghost module makes it easier to deploy into UAVs and increase the speed of prediction, but these are all at the expense of accuracy.

#### 4.2.3. Influence of Small Object Detection Network

To solve decreasing accuracy of the Ghost module, we chose to change the network structure to increase the accuracy. Table 1 shows that the average accuracy increases by 0.1% in the evaluation of mAP0.5 and 2.1% in the evaluation of mAP0.5:0.95 compared with the YOLOv5s + Ghost module. For defect, the mAP0.5 increases by 0.2%, and mAP0.5:0.95 increases by 4.7%. Although the deploy performance is slightly inferior to YOLOv5s + Ghost module, it is much superior to YOLOv5s and has significant improvement in small defect detection. accuracy decreases by 0.3% in the evaluation of mAP0.5 and 1.3% in the evaluation of mAP0.5:0.95 in Table 1. For insulators, the mAP0.5 decreases by 0.3%, and mAP0.5:0.95 decreases by 1.1%. For defect, the mAP0.5 decreases by 0.2%, and mAP0.5:0.95 decreases by 1.5%. Ghost module makes it easier to deploy into UAVs and increase the speed of prediction, but these are all at the expense of accuracy.

#### 4.2.4. Influence of CBAM

CBAM was proposed for our network structure to improve the model accuracy further. Table 1 shows the average accuracy increases by 0.1% in the evaluation of mAP0.5 and 0.5% in the evaluation of mAP0.5:0.95 compared with YOLOv5s + Ghost module + small object detection network. For insulators, the mAP0.5 increases by 0.1%, and mAP0.5:0.95 increases by 0.9%.

### 4.3. Comparison with Different Methods

We select some classic models and state-of-the-art models for comparison to verify the accuracy and performance of our model, which include Faster R-CNN, YOLOv3, and YOLOv4. All of the networks were trained on our proposed dataset. Table 3 shows the results of network performance.

As shown in Table 3, the classic method, faster R-CNN, has poor performance on insulator and defect detection and UAV deployment. The speed of YOLOv3 and YOLOv4 is slightly faster than ours, but the mean average precision is much lower than our model. Our model maintains the high accuracy of YOLOv5s and greatly improves the capability of small defect detection. Besides, we halve its parameters and model size to make it easier to deploy to embedded systems. The experiment results show that our model has better superiority and applicability through comparison.

As shown in Figure 9, we compared the original YOLOv5s network and our network. In Figure 9A, YOLOv5s cannot detect the insulator in the corner of the image, but our proposed network can detect it correctly. In Figure 9B–D, we selected the pictures under foggy, sunny, and normal conditions. YOLOv5s network and our network can both correctly detect insulators and defects, but the accuracy of our network is much higher than the YOLOv5s network. In Figure 9E, the YOLOv5s network had a wrong detection of insulators, and it cannot detect the defect; our model can detect the insulators and defects with high accuracy. The comparison results show the superiority of our network.

Through the experiments, Faster R-CNN has poor robustness and a large number of parameters, so we selected the YOLO framework as our base framework. YOLOv3 and YOLOv4 have low detection rates on insulators and defects, so we select the state-of-the-art YOLOv5 network. We applied the Ghost module to reduce the model size and parameters of YOLOv5. Besides, to improve the detection rate of small defects, we proposed a small object detection network and applied it to the neck of the YOLOv5. In addition, to further improve the accuracy of our method, we introduced CBAM to the backbone of the network.

## 5. Conclusions

This article proposes a lightweight insulator and defect detection network based on the improved YOLOv5, which meets the requirement of industrial deployment. The conclusions are as follows:Ghost module is introduced to the network structure of YOLOv5, which greatly decreases the parameters and FLOPs of the network, reduces the model size by half, and maintains high detection speed.Applying the CBAM module can increase insulator detection accuracy with only a slight increase in model size and computation cost.The network changes for small object detection make it easier to detect small defects and significantly increase the mean average precision of defect detection.

By comparing experiment results, our model has better predictive and deploy performance than classic and state-of-the-art models. The average mAP0.5 can reach 99.4%, and the average mAP0.5:0.95 can reach 91.7%. Moreover, the speed of detection is 10.9 ms/image, which can fully meet the real-time requirement. Moreover, the parameters and model size is reduced to 3,807,372 and 8.79 M. Its small model size and low computation cost make it easier to deploy to embedded devices such as UAVs and reduce industrial costs.

## Figures and Tables

**Figure 1 sensors-23-05249-f001:**
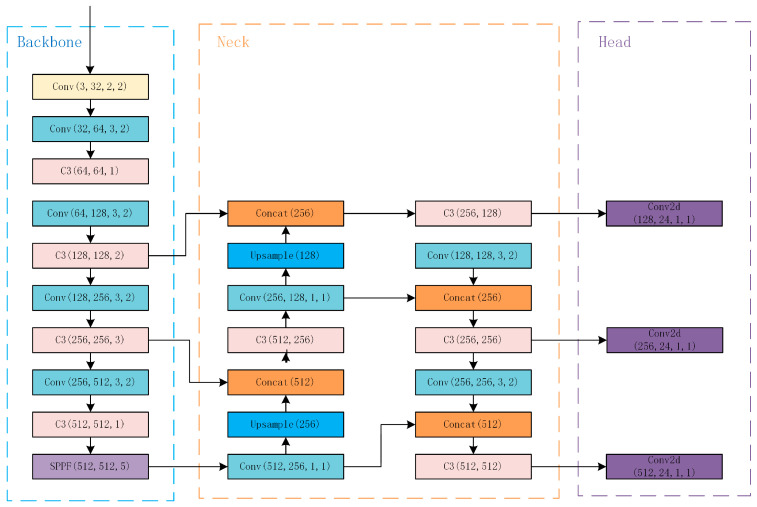
The network structure of YOLOv5.

**Figure 2 sensors-23-05249-f002:**
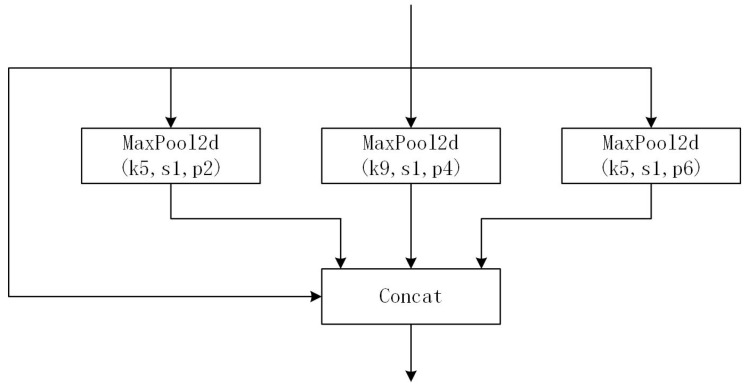
The architecture of the SPP layer.

**Figure 3 sensors-23-05249-f003:**
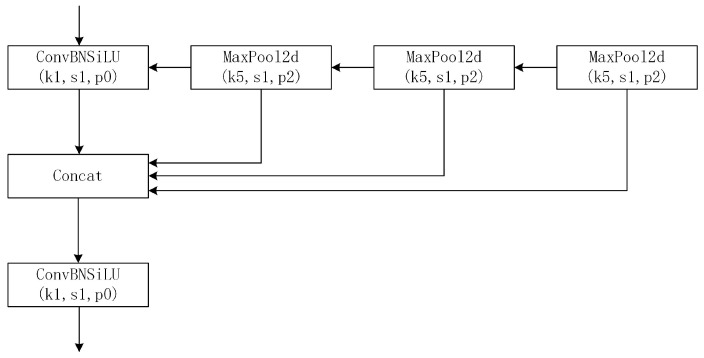
The architecture of the SPPF layer.

**Figure 4 sensors-23-05249-f004:**
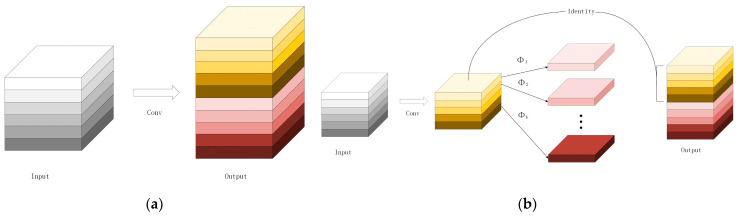
The working process of (**a**) the original convolutional layer and (**b**) the Ghost module.

**Figure 5 sensors-23-05249-f005:**
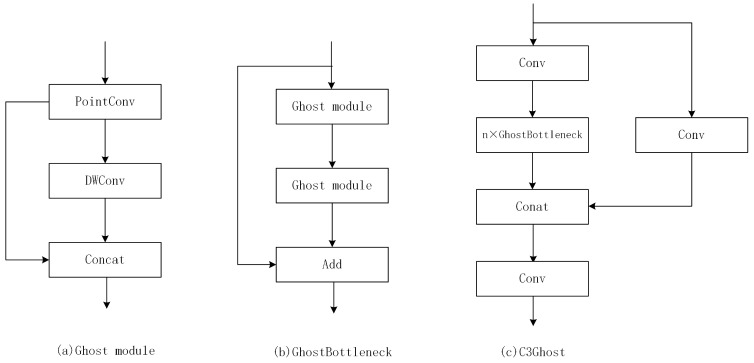
The architecture of Ghost module, GhostBottleneck and C3Ghost.

**Figure 6 sensors-23-05249-f006:**
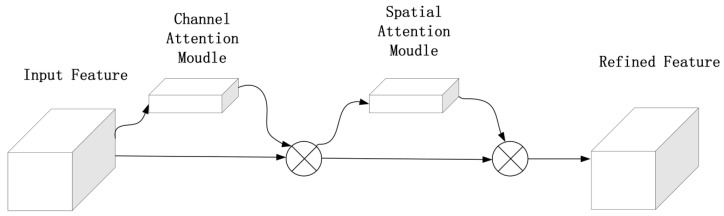
The CBAM block.

**Figure 7 sensors-23-05249-f007:**
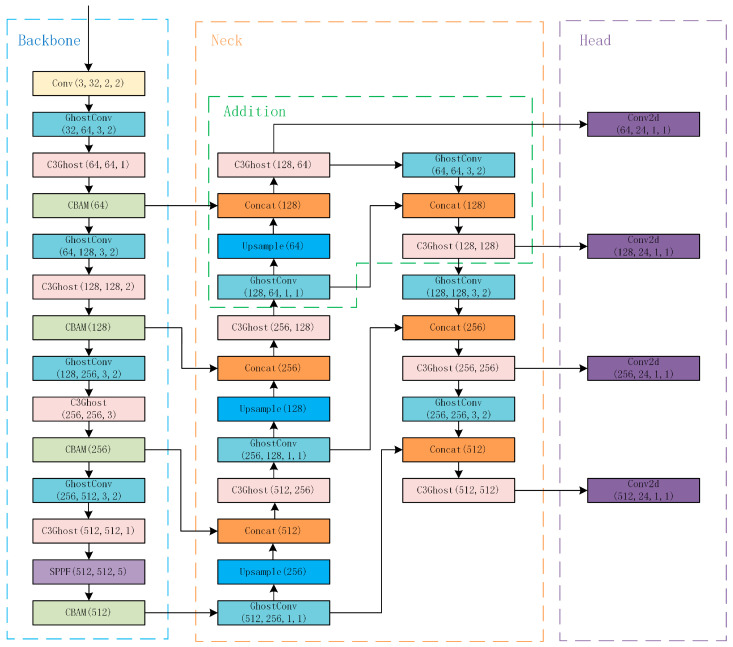
The overall network structure.

**Figure 8 sensors-23-05249-f008:**
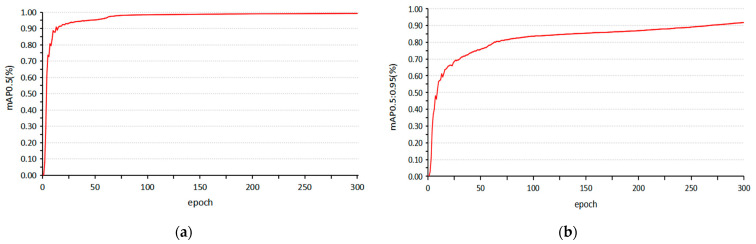
(**a**) The training curve of our model in mAP0.5 (**b**) The training curve of our model in mAP0.5:0.95.

**Figure 9 sensors-23-05249-f009:**
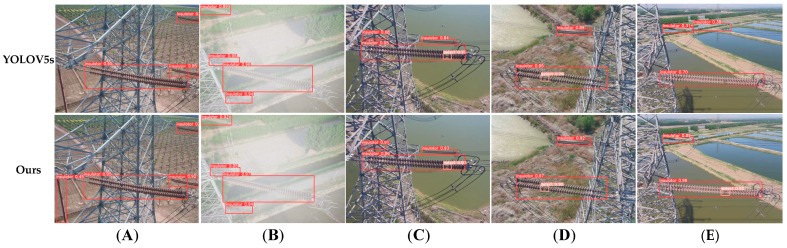
Insulator and defect detection using YOLOv5s and our model. (**A**) The different insulator detection rates of YOLOv5s and our model. (**B**–**D**) The detection rates in different weathers. (**E**) The different insulator and defect detection rates of YOLOv5s and our model.

**Table 1 sensors-23-05249-t001:** Detection results of insulators and defects in different network structures.

Model	Classes	Precision	Recall	mAP0.5	mAP0.5:0.95
YOLOv5s	Average	99.4%	99.3%	99.5%	90.4%
	Insulator	99.2%	98.8%	99.4%	93.1%
	Defect	99.6%	99.7%	99.5%	87.6%
YOLOv5s + Ghost module	Average	98.4%	98.5%	99.2%	89.1%
	Insulator	97.2%	97.4%	99.1%	92.0%
	Defect	99.6%	99.6%	99.3%	86.1%
YOLOv5s + Ghost module + small object detection network	Average	98.6%	98.9%	99.3%	91.2%
	Insulator	97.2%	98.1%	99.2%	91.6%
	Defect	99.9%	99.7%	99.5%	90.8%
YOLOv5s + Ghost module + small object detection network + CBAM	Average	98.7%	98.9%	99.4%	91.7%
	Insulator	97.9%	98.0%	99.3%	92.5%
	Defect	99.6%	99.7%	99.5%	90.8%

**Table 2 sensors-23-05249-t002:** Deploy performance in different network structures.

Model	Parameters	FLOPs (G)	Size (M)	Speed-GPU (ms/Image)
YOLOv5s	7,025,025	16.0	13.72	9.5
YOLOv5s + Ghost module	3,687,239	8.2	7.44	9.3
YOLOv5s + Ghost module + small object detection network	3,763,460	9.8	8.69	10.5
YOLOv5s + Ghost module + small object detection network + CBAM	3,807,372	9.9	8.79	10.9

**Table 3 sensors-23-05249-t003:** Comparison of different network performances.

Model	mAP0.5	mAP0.5:0.95	Parameters	FLOPs (G)	Size (M)	Speed-GPU (ms/Image)
Faster R-CNN	97.2%	77.8%	19,546,215	7.8	74.25	8.8
YOLOv3	98.8%	79.7%	8,654,686	12.8	16.68	8.7
YOLOv4	99.2%	83.5%	8,787,543	16.5	11.34	9.1
YOLOv5s	99.5%	90.4%	7,025,025	16.0	13.72	9.5
Ours	99.4%	91.7%	3,807,372	9.9	8.79	10.9

## Data Availability

Not applicable.

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
