# Peer review of "A Light-Weight Network for Small Insulator and Defect Detection Using UAV Imaging Based on Improved YOLOv5"

_sensors, 2023, doi:10.3390/s23115249_

Round 1
Reviewer 1 Report
The manuscript is written with clear understanding of the project addressed. However, there are major concerns that need to be addressed to enhance the quality of the manuscript. My specific comments are as follows:
Introduction:
L39:” We can divide object detection methods into two types [5].” Specify the types.
Based on your objectives, please compare how your study is different from those that have already been published.
Spell out the acronym for the first time you mentioned in the text.
Results and Discussion:
Add accuracy in Table 1.
Relate your results with existing literatures to support your findings. Instead of mentioning the results, the authors should justify/explain the findings
General comments:
Please check the reference styles and grammar of the manuscript.
Minor editing of English language required.
Author Response
Thanks for your comments concering our manuscript entitied "A Light-weight Network for Small Insulator and Defect Detection Using UAV Imaging Based on Improved YOLOv5". Those comments are valuable and helpful. We have read through comments carefully and have made corrections.
Q1:L39:” We can divide object detection methods into two types [5].” Specify the types.
Response: We explanded these two types in 3rd and 4th paragraph.
Q2:Based on your objectives, please compare how your study is different from those that have already been published.
Response: We added 8th paragraph to compare the methods have been published.
Q3:Spell out the acronym for the first time you mentioned in the text.
Response: Unmanned aerial vehicles (UAVs) was mentioned in abstract.
Q4:Add accuracy in Table 1.
Response: We used mean average precision (mAP) as the measurement indicators, it is more suitable in object detection methods than accuracy.
Q5: Relate your results with existing literatures to support your findings. Instead of mentioning the results, the authors should justify/explain the findings.
Response: We added last paragraph for comparision.
Q6: Please check the reference styles and grammar of the manuscript.
Response: All references were in EndNote MDPI style, and we have corrected most of our grammar mistakes.
Reviewer 2 Report
The authors have proposed a light-weight network for insulator and defect detection. In this network, Ghost module is introduced into YOLOv5 backbone and neck to reduce the parameters and for small object detection, a small object defect detection network is added. Finally a convolutional block attention module (CBAM0 is also added to improve accuracy.
Basically additional modules are increased to YOLOv5.
Simulation results are shown along with comparison with YOLOv5.
I have a few suggestions to make the paper better:
1. Please add the accuracy plot during training and testing.
2. How many iterations are required by the proposed work to achieve this accuracy. Mention the time taken for training.
3. Please expand mAP0.5 and mAP0.5:0.95 during the first usage for clarity.
4. First para of Introduction, 3 rd line should be interferences.
5. A lot of FLOPS. Please change to higher number of FLOPS
6. After adding three modules to YOLOv, how you claim light weight algorithm as computations would only increase.
7. Please check the grammar of the sentence, "As one-stage algorithm, compare to the two-stage algorithm, is faster and more suitable 83 for real-time insulator defect detection in UAVs."
8. Section 2.3 first line, check the grammar of the sentence "The head is three detection layers"
9. Please explain the fig 4(b) clearly
10. Delete the line in page 6, "This section may be divided by subheadings. It should provide a concise and precise description of the experimental results, their interpretation, as well as the experimental conclusions that can be drawn"
11. The title Experimental Introduction can be removed and instead 4.1.1 can be made 4.1. Change the section numbers after this accordingly.
12. in data set description, three data set are used. Change two to three. It is grammatical wrong to start the sentence with And. Correct the paragraph.
13. Correct the sentence, "All experiments are in Linux operating system," with correct english grammar.
14. Expand IoU.
15. How do you define Weight and how was Weight of the algorithms calculated.
16. Figure 8 shows the Insulator and defect detection using YOLOv5s and our model under different conditions. Can you add a table a comparison table indicating the what the different conditions are and the performance indicators that would vary for these different conditions.
There are many grammatical errors in the manuscript . I have found many and indicated to you.
Please check once again thoroughly
Author Response
Thanks for your comments concering our manuscript entitied "A Light-weight Network for Small Insulator and Defect Detection Using UAV Imaging Based on Improved YOLOv5". Those comments are valuable and helpful. We have read through comments carefully and have made corrections.
Q1: Please add the accuracy plot during training and testing.
Response: We added mAP0.5 and mAP0.5:0.95 plot in 4.2.1.
Q2: How many iterations are required by the proposed work to achieve this accuracy. Mention the time taken for training.
Response: We trained 300 epochs and we added it in 4.2.1.
Q3: Please expand mAP0.5 and mAP0.5:0.95 during the first usage for clarity.
Response: We expanded it in 4.1, it could not be expanded without expanding precision, recall, and IoU. So we prefer to put it on 4.1.
Q4: First para of Introduction, 3 rd line should be interferences.
Response: We have corrected it in new version.
Q5: A lot of FLOPS. Please change to higher number of FLOPS
Response: We have corrected it in new version.
Q6: After adding three modules to YOLOv, how you claim light weight algorithm as computations would only increase.
Response: We reduced the model size, parameters and FLOPs. Because of operator transformation, the speed on GPU is a little slower than YOLOv5.
Q7: Please check the grammar of the sentence, "As one-stage algorithm, compare to the two-stage algorithm, is faster and more suitable 83 for real-time insulator defect detection in UAVs."
Response: We have corrected it in new version.
Q8: Section 2.3 first line, check the grammar of the sentence "The head is three detection layers"
Response: We have corrected it in new version.
Q9: Please explain the fig 4(b) clearly
Response: We have explained it in 3.2.
Q10: Delete the line in page 6, "This section may be divided by subheadings. It should provide a concise and precise description of the experimental results, their interpretation, as well as the experimental conclusions that can be drawn"
Response: We have corrected it in new version.
Q11: The title Experimental Introduction can be removed and instead 4.1.1 can be made 4.1. Change the section numbers after this accordingly.
Response: We have corrected it in new version.
Q12: in data set description, three data set are used. Change two to three. It is grammatical wrong to start the sentence with And. Correct the paragraph.
Response: We have corrected it in new version.
Q13: Correct the sentence, "All experiments are in Linux operating system," with correct english grammar.
Response: We have corrected it in new version.
Q14: Expand IoU.
Response: We expanded it in 4.1
Q15: How do you define Weight and how was Weight of the algorithms calculated.
Response: We have changed model weight to model size.
Q16: Figure 8 shows the Insulator and defect detection using YOLOv5s and our model under different conditions. Can you add a table a comparison table indicating the what the different conditions are and the performance indicators that would vary for these different conditions.
Response: We explained the conditions in Figure 8, and its very hard to caculate mAP in each condition.
Reviewer 3 Report
The paper is 11 and a half pages long. About 3 pages are illustrations and 1 page is references. The remainder of the content is rather short for journal standards.
Subsections 2.3 and 4.1.2 are one sentence long each. The figure (Fig. 1) provided in that section is not explained.
The textual contents of the boxes in Figures 1, 2, 3, and 7 are not explained.
Figure 8 is not explained. I cannot understand what I should be looking at or comparing in this figure.
The paper is difficult to follow for readers not accustomed to the details of convolutional methods. The paper could benefit from definitions and explanations of key terms in this topic.
The terms mAP0.5 and mAP0.5:0.95 are not explained.
The experiments compare different applications of the YOLO framework but do not compare with other methods in the literature, regarding the same topic.
Though the manuscript is comprehensible, there are severe grammatical errors that need to be corrected.
Author Response
Thanks for your comments concering our manuscript entitied "A Light-weight Network for Small Insulator and Defect Detection Using UAV Imaging Based on Improved YOLOv5". Those comments are valuable and helpful. We have read through comments carefully and have made corrections.
Q1: The paper is 11 and a half pages long. About 3 pages are illustrations and 1 page is references. The remainder of the content is rather short for journal standards.
Response: We have expanded some figures and concepts to meet the journal standards.
Q2: Subsections 2.3 and 4.1.2 are one sentence long each. The figure (Fig. 1) provided in that section is not explained.
Response: We have changed the 4.1 chapter and expanded figure 1 in 2.3.
Q3: The textual contents of the boxes in Figures 1, 2, 3, and 7 are not explained.
Response: We have explained Figures 1,2,3, and 7 in new version and expanded 4(b) for readers.
Q4: The paper is difficult to follow for readers not accustomed to the details of convolutional methods. The paper could benefit from definitions and explanations of key terms in this topic.
Response: We have explained most Figures and the key terms in 4.1 in new version.
Q5: The terms mAP0.5 and mAP0.5:0.95 are not explained.
Response: We have explained mAP0.5 and mAP0.5:0.95 in 4.1 in new version.
Q6: The experiments compare different applications of the YOLO framework but do not compare with other methods in the literature, regarding the same topic.
Response: The most mainstream object detection method is YOLO framework. Besides, we compared Faster-RCNN in the literature. Some other improved algorithms are not open source, we can only get the mAP in their dataset. Because of the number of images and the and annotations are different, their mAP are not suitable for our dataset. Besides, we cannot get their parameters, model size, and FLOPs. So, we chose to compare the open source methods Faster-RCNN and YOLO family on our dataset.
Round 2
Reviewer 3 Report
The authors have dully responded to the comments of the reviewers and the paper merits publication.
Some syntactical errors still remain. I am sure that a final proofreading should fix them.